# INSTAX3D: CREATING 3D PORTRAIT FROM A SINGLE-VIEW IMAGE IN MINUTES

## ABSTRACT

We study single-view 3D portrait creation, specifically producing a full-head 3D portrait from a single headshot. This problem faces two challenges: 1) the 2D image-based personalization methods lack comprehensive 3D awareness due to the scarcity of multi-view 2D images or 3D assets in the training data, and 2) the score distillation sampling optimization methods usually take hours to produce a single 3D asset, making the process quite time-consuming. To overcome these limitations, we propose Instax3D, a generative Gaussian Splatting model with a video diffusion prior for rapid 3D portrait creation. We formulate the 3D portrait creation problem as a "generation and construction" process. Specifically, Instax3D first synthesizes a consecutive video sequence using a finetuned video diffusion model, capitalizing on inherent diversity and multi-view knowledge from the massive video data. Subsequently, Instax3D reconstructs the 3D portrait with a multi-view FLAME-based Gaussian splatting representation from the generated video frames, structurally guided by an expressive 3D parametric model. Notably, given a reference headshot image, Instax3D can generate a 3D portrait in just 10 minutes and render it at 40 FPS. This represents a $10\times$ improvement over previous mainstream optimization-based methods, which can take between one to two hours. Our project page: Instax3D Webpage.

## 1 INTRODUCTION

Single-view 3D portrait creation is designed to model a full-head 3D portrait from only one headshot. Creating 3D assets of human heads has been a long-standing problem in computer vision and graphics. A wide range of downstream applications have emerged in various fields, including immersive telepresence, digital human avatars, virtual and augmented reality, the gaming industry, and movie production. Thanks to the successful advancements in generative models (Karras et al., 2020; Rombach et al., 2021; Song et al., 2021b; Ho et al., 2022), recent developments in large-scale diffusion models have paved the way for creating photorealistic and lifelike 3D content. These breakthroughs greatly expanded the potential for 3D portrait generation. In practical applications, an ideal generative model for 3D portrait creation should meet the following criteria: (1) *Strong 3D Prior and Geometry Awareness*: The model should be capable of effectively conceptualizing 3D geometry and reconstructing the appearance of a complete 3D head portrait from just a single reference photo. (2) *Generalizability and Identity Preservation*: It is essential for the generative model to accurately capture and maintain the identity features, ensuring it adapts well to new characters. (3) *Rapid Training Capability*: It would be preferable for the model training to take only a few minutes.

Early methods use 3D-GAN inversion (Bhattarai et al., 2024; Wu et al., 2023) to find the corresponding latent code for the reference headshot photo. These methods (Bhattarai et al., 2024; Wu et al., 2023) first train 3D-aware face generators (Chan et al., 2022a; Sun et al., 2023a) on that consist of near-frontal face images, *e.g.*, FFHQ (Karras et al., 2019) and CelebAHQ (Karras et al., 2018). Then, the pivotal tuning inversion (PTI) (Roich et al., 2022) techniques are used to refine the latent code and adjust the generator weights. Due to the scarcity and limited diversity of training images, *i.e.*, identities and poses, these methods often produce collapsed results under large pose variations and struggles to scale to in-the-wild images.

Recently, diffusion models (Song et al., 2021b) have achieved superior performance in generating portraits across 2D images and 3D shapes. Given a reference image, 2D portrait image genera-

tion models (Liang et al., 2024; Ye et al., 2023; Wang et al., 2024) employ a pre-trained image encoder (Radford et al., 2021) to project the face image into the feature space, and then integrate the features into the denoising U-Net with the cross-attention mechanism. The image condition design can preserve the identity of the reference image effectively. For 3D generation, optimization-based methods (Qian et al., 2024; Tang et al., 2023; Sun et al., 2023b; Xu et al., 2023) utilize score distillation sampling (SDS) (Poole et al., 2022) to produce 3D assets by distilling the image-conditioned 2D diffusion prior into 3D representation. However, these methods have notable drawbacks. The 2D image-based personalization methods (known as 2D-lifting) typically rely on single-view 2D image diffusion models (Rombach et al., 2021), which do not incorporate 3D spatial awareness or information from multiple perspectives. Consequently, these human portrait generation methods (Chang et al., 2023; Xu et al., 2024; Liang et al., 2024; Ye et al., 2023; Wang et al., 2024) are limited to frontal views with small pose variations. SDS-based methods (Qian et al., 2024; Wang & Shi, 2023) often require hours to optimize a single 3D asset and struggle with identity preservation, making it fail to satisfy the requirements for practical applications. This highlights the increasing demand for more advanced solutions capable of generating full-head identity-preserving 3D portraits within a short time.

To address these limitations, we propose Instax3D, a generative Gaussian Splatting model with video diffusion prior for fast 3D portrait creation. Different from techniques (Qian et al., 2024) utilizing 2D diffusion models (Rombach et al., 2021) as supervision, we build Instax3D upon the video diffusion model trained on massive videos, and thus can inherently learn a generalizable 3D prior. In particular, we formulate the 3D portrait creation problem as a "generation and construction" process. We finetune a video diffusion model (Voleti et al., 2024) to imagine and hallucinate the shape and appearance of a human full-head from only one reference photograph. (1) In the generation stage, Instax3D deploy the fine-tuned video diffusion model to synthesize a consecutive multi-view video sequence, leveraging the generalization and multi-view consistency from the generative video prior. (2) In the construction stage, we directly reconstruct the corresponding 3D portrait from the video frames. Specifically, we design an efficient 3D representation by incorporating 3D Gaussian Splatting (Kerbl et al., 2023) with the expressive 3D morphable face model FLAME (Li et al., 2017). The former enables Instax3D to reduce optimization time to just a few minutes and enhance the rendering speed, while the latter allows further acceleration of the training process by fully harnessing the rich geometry priors as explicit structural guidance.

In summary, the contributions of this paper are three-fold:

1. We introduce Instax3D, a generative framework that produces photorealistic and identity-preserving 3D portraits from a single-view photograph within a few minutes.

2. We propose a novel solution to 3D portrait generation, which formulates the problem as a "generation and construction" process, harnessing multi-view knowledge from video diffusion priors and the rapid convergence capability of Gaussian splatting representation.

3. We conduct quantitative and qualitative evaluations of Instax3D, demonstrating its superiority over previous state-of-the-art methods.

## 2 RELATED WORK AND PRELIMINARIES

### 2.1 RELATED WORKS

**Image-guided 3D Content Generation.** The successful advancement in diffusion models brings the dawn of possibilities for 3D generation. Recent works have explored ways to integrate diffusion models into image-guided 3D content generation. One straightforward approach involves first estimating coarse geometric properties, such as depth and normals, from generated 2D images, and then fine-tuning the 2D diffusion model for novel view generation using the subject-driven technique DreamBooth (Ruiz et al., 2023). Given a reference image, Magic123 (Qian et al., 2024) initially constructs a coarse geometry with neural radiance fields, and then deploys a differentiable and memory-efficient rasterizer to optimize a high-resolution mesh with detailed features. HiFi-123 (Yu et al., 2023) incorporates a novel view enhancement along with a reference-guided state distillation loss. Some methods embrace a 2D-lifting paradigm by leveraging an image caption model to generate a text prompt from the given image and performing score distillation sampling (SDS) (Poole et al., 2022) for 3D generation. Tang et al. (2023) consider the reference image as the ground truth of the

frontal view and leverage the diffusion prior from other views. Recently, some researchers have also explored the fast reconstruction implementation in a feed-forward manner. Long et al. (2024) introduce a cross-domain diffusion model that can produce multi-view RGB images and normal images simultaneously. The proposed multi-view cross-domain attention mechanism of Wonder3D (Long et al., 2024) facilitates cross-view and cross-modalities information exchange and allows for high-quality geometry generation. Hong et al. (2024) introduce LRM, a scalable encoder-decoder transformer framework for 3D object generation from the input image. Wang & Shi (2023) extend MVDream (Shi et al., 2024) to an image-conditioned multi-view diffusion model that takes image-prompt as input. After passing the CLIP encoding to both local and global controllers (Wang & Shi, 2023), the output image features are then inserted into cross-attention layers to guide the 3D generation. Despite the efficiency and fast reconstruction speed, these methods are restricted to some simple objects due to the sparse input views and insufficient cross-view consistency.

**Portraits Generation.** Creating photo-realistic human portraits from user commands such as text descriptions, target poses, and reference images plays an important role in real-world applications. Early approaches (Zhu et al., 2017; Siarohin et al., 2019; Yang et al., 2021) adopt variational autoencoder (VAE) (Goodfellow et al., 2020) or conditioned generative adversarial networks (GAN) (Goodfellow et al., 2020) to guide the image synthesis. Liang et al. (2024) propose to use both a fine local and a coarse global encoder to project the reference photograph to an aligned identity feature into the latent space. To achieve fine-grained control for the human head, Liang et al. (2024) introduce the facial prior obtained from a 3D Face reconstruction module as conditional guidance. Ye et al. (2023) design a lightweight adapter that contains an image decoder and a decoupled cross-attention module. By replacing the CLIP image embedding with face ID embedding extracted from a face recognition model, IP-adapter (Ye et al., 2023) can generalize to face portrait generation with LoRA (Hu et al., 2022) fine-tuning technique to improve the face ID consistency. Imposing semantic and spatial conditions with an IdentityNet module, InstantID (Wang et al., 2024) achieves impressive results in personalized image synthesis while maintaining face fidelity. However, all the above methods are limited to small viewpoint variation scenarios, making it hard to create full-head $360°$ free-view portrait generation.

## 2.2 PRELIMINARIES

**3D Gaussian Splating.** In the heart of 3D Gaussian Splatting (Kerbl et al., 2023) is a real-time radiance field (Mildenhall et al., 2020) reconstruction process that utilizes an efficient and expressive point-based explicit representation and a fast differentiable rasterizer for 3D Gaussians. The former is a differentiable volumetric representation that can be rasterized by projecting to 2D image space with standard $\alpha$-blending, while the latter supports fast anisotropic splatting when combined with a visibility-ordering rendering algorithm, thus achieving real-time rendering and accelerating the optimization process.

3D Gaussian Splatting (Kerbl et al., 2023) model the scene with a dense set of anisotropic Gaussian kernels: $\{G_i\} = \{\mu_i, \alpha_i, \Sigma_i, c_i\}$, where $G_i$ is the $i$-th kernel, $\mu_i \in \mathbb{R}^3$ is the center position, $\alpha_i \in \mathbb{R}$ is the opacity, $\Sigma_i$ is the anisotropic 3D covariance matrix, and $c_i \in \mathbb{R}^3$ is the color represented by spherical harmonics for view-dependent appearance. In the world space, the Gaussian splats are defined with mean $\mu_i$ and covariance $\Sigma_i$ :

$$G_i(x) = e^{-\frac{1}{2}(x-\mu_i)^T \Sigma_i^{-1}(x-\mu_i)},\tag{1}$$

where $x$ is the coordinate of the queried point, and the covariance matrix $\Sigma_i$ is factorized into a diagonal scaling matrix $S_i$ and an orthogonal rotation matrix $R$ to guarantee the semi-definite property:

$$\Sigma_i = R_i S_i S_i^T R_i^T.\tag{2}$$

When rendering, 3D Gaussian splats $G_i$ can be easily projected onto the 2D image plane as 2D Gaussians $G_i^{2D}$. The 2D covariance matrix $\Sigma_i^{2D}$ corresponding to $G_i^{2D}$ is calculated with:

$$\Sigma_i' = JV\Sigma_i V^T J^T, \qquad \Sigma_i^{2D} = \Sigma_i'[:2,:2],\tag{3}$$

where $V$ is the world-to-camera matrix, $\Sigma_i$ is the 3D covariance matrix, and $J$ is the Jacobian by approximating the affine of the projective transformation. The 2D splats $G_i^{2D}$ with standard

$\alpha$-blending for fast rendering:

$$c(x) = \sum_{i=1}^{N} c_i \alpha_i G_i^{2D}(x) \prod_{j=1}^{i-1}(1 - \alpha_j G_j^{2D}(x)), \qquad (4)$$

where $N$ denotes the number of the sorted of 3D Gaussians in this tile, $c_i$ is the spherical harmonics coefficient, and $\alpha_i$ is the opacity.

**FLAME.** FLAME (Li et al., 2017) model (Faces Learned with an Articulated Model) adapts the Skinned Multi-Person Linear model (Loper et al., 2015) formulation to 3D human head scenarios. FLAME (Li et al., 2017) is a statistical parametric human model that can represent a wide range of face identity shapes, poses, and expressions. Given a set of parameter $p = (\boldsymbol{\beta}, \boldsymbol{\theta}, \boldsymbol{\psi})$ that includes shape $\boldsymbol{\beta} \in \mathbb{R}^{|\boldsymbol{\beta}|}$, pose $\boldsymbol{\theta} \in \mathbb{R}^{3k+3}$(with $k = 4$ joints for jaw, neck, and eye gaze), and expression $\boldsymbol{\psi} \in \mathbb{R}^{|\boldsymbol{\psi}|}$, FLAME defines a deformable template mesh $M(\boldsymbol{\beta}, \boldsymbol{\theta}, \boldsymbol{\psi})$ with 5023 vertices and 9976 faces. In this work, the FLAME (Li et al., 2017) mesh can provide a coarse geometric proxy for the synthesized 3D portrait.

## 3 METHODOLOGY

Our goal is to generate a 3D full-head portrait from only one single-view heatshot image. To achieve this, we propose Instax3D for efficient 3D Gaussian head creation, leveraging the video diffusion model for multi-view generation and using structural prior of 3D head geometry template for construction. Notably, with careful design, Instax3D can accelerate the 3D portrait creation time within 10 minutes while maintaining high-fidelity identity preservation ability.

### 3.1 OVERVIEW

We formulate the 3D portrait creation as a "generation and construction" process. The core idea is to first use a finetuned video diffusion model to generate a set of consecutive multi-view video frames from the reference image, and then construct the 3D head by distilling the underlying geometric prior from the generated video. The pipeline of the proposed method is illustrated in Figure 1. In Section 3.2, we first introduce how to adapt the video diffusion model to the 3D head scenarios for multi-view portrait video generation. Then, in Section 3.3, we elaborate on how to construct the 3D head with a FLAME-based Gaussian representation from the generated video frames.

### 3.2 GENERATION

We build the multi-view portrait generation module by adapting an image-to-video generator network, *i.e.*, Stable Video 3D (SV3D) (Voleti et al., 2024), to the human head scenarios. Stable Video 3D (SV3D) (Voleti et al., 2024) is an image-conditioned video diffusion model that supports generating orbital video with explicit camera control. Given a single-view reference image $\mathcal{I} \in \mathbb{R}^{3 \times H \times W}$ as the first frame, SV3D generate a multi-view video sequence $\mathcal{S} \in \mathbb{R}^{K \times 3 \times H \times W}$ with the camera pose trajectory $\boldsymbol{\pi} \in \mathbb{R}^{K \times 2} = \{(e_i, a_i)\}_{i=1}^{K}$. Here $K$ denotes the number of video frames, while $e_i$ and $a_i$ are the specified elevation and azimuth angles. In our setting, we aim to generate consecutive headshot video clips to provide sufficient multi-view supervision and cross-view consistency for the 3D construction stage. Therefore, we finetune the pre-trained SV3D (Voleti et al., 2024) model to generate a specific type of 360-degree selfie video, where the character rotates the head in front of the camera in a turn-table-like fashion.

**Finetune Portrait Video Generator.** The denoising U-Net is composed by the encoder, middle, and decoder blocks, and each block contains multiple basic units at different resolutions. For each unit, there is one residual block with 3D convolution layers, followed by a spatial transformer and a temporal transformer block in a sequential manner. To capture the intricate appearance feature and preserve the identity information, we deploy a dual-way paradigm to merge the image feature of the reference portrait into the denoising UNet. As shown in the Figure 1 (a), the reference image $\mathcal{I}$ is projected to two kinds of features with a pre-trained frozen SVD (Blattmann et al., 2023a) VAE encoder and a CLIP (Radford et al., 2021) encoder, yielding a VAE embedding $y_{vae}$ and a CLIP feature $y_{vae}$ respectively. Given the original latent noise $z_t$ at the denoising timestep $t$, the VAE embedding $y_{vae}$ is directly injected into the latent space via concatenation: $z_t' = [z_t, y_{vae}]$. The

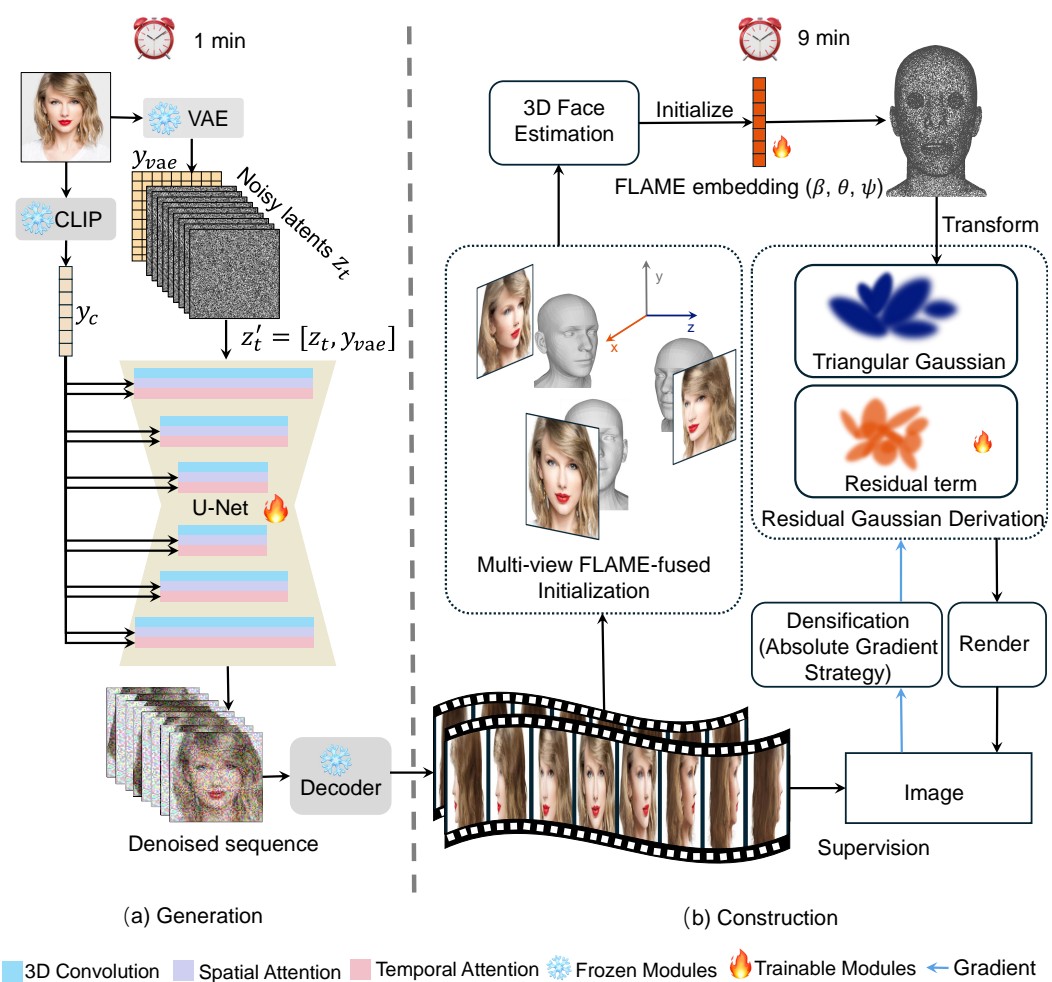

Figure 1: The overview of Instax3D. (a) In the generation stage, Instax3D extend a single-view image to a set of consecutive multi-view video frames with a finetuned video diffusion model. (b) In the construction stage, Instax3D constructs the 3D portrait with Gaussian splitting representation. To facilitate the reconstruction process, we adopt several effective strategies, like multi-view FLAME-based initialization, residual gaussian derivation, and absolute gradient strategy.

updated input latent $z'_t$ is then passed into the first 3D convolution residual block of the U-Net, resulting in a good initialization for both intra- and inter-frame the space and time dimensions. Then a spatial transformer layer is employed to model the spatial-structured relationship by treating the video latent sequence as a batch of independent image features, while the subsequent temporal transformer block performs temporal fixing with a self-attention sub-layer along the temporal dimension. Suppose $z''$ is the input feature of a transformed layer, the clip feature $y_c$ is integrated into the transformer blocks as the image prompt. The cross-attention module attends from the clip feature $y_c$ to the latent features $z''$, conditioning on both the spatial and temporal attention blocks:

$$\text{Attention}(Q, K, V, y_c) = \text{softmax}(\frac{QK^T}{\sqrt{d}}) \cdot V, \tag{5}$$

where $Q = W_Q \cdot z''$, $K = W_K \cdot y_c$, $V = W_V \cdot y_c$ are the query, key, and values matrices, respectively. Lastly, the output features will become the input to the next residual or transformer block.

**Camera Invariant Modulation.** Existing works typically incorporate camera embeddings (Hong et al., 2024; Chan et al., 2022b; Blattmann et al., 2023b) as a strong prior knowledge to guide the generative model to produce similar results with the same condition. In practice, we find the camera-aware condition can cause collapsed and distorted results in the novel views. In our 3D portrait

creation setting, the camera parameters are estimated by the off-the-shelf 3D human detector. Due to the domain gap of camera distribution between the training data and the testing data, even slight perturbation caused by the inaccurate estimation of the camera parameters can lead to de-generated results. Therefore, we introduce the camera-invariant modulation with zero camera embedding for more robust generation results.

### 3.3 CONSTRUCTION

In the generation stage, we build the multi-view FLAME-based 3D Gaussian splitting construction from the video sequence obtained from the generation stage.

**Multi-view FLAME-fused Initialization.** We supplement structural knowledge to get a good initialization to accelerate the training process and ease the 3D GS learning. Previous studies derive the center of 3D Gaussian kernels as the point clouds by using either multi-view Structure-from-Motion (SfM) points with COLMAP techniques (Snavely et al., 2006) or the predicted mesh from the single-view reconstruction method (Garau et al., 2021; Daněček et al., 2022). However, the SFM reconstruction method (Snavely et al., 2006) typically requires many input images with different viewpoints to achieve reasonable results and has a higher requirement of the pixel correspondence between the input images, while the single-view estimated result is inaccurate due to the 2D-to-3D ambiguity. In this work, we combine two methods by estimating FLAME mesh from multiple views and fusing them all to obtain a well-initialized result. Given the generated multi-view video sequence $\mathcal{S} \in \mathbb{R}^{K \times 3 \times H \times W}$, we select 3 frames (including the reference image and two neighbor frames) and use an off-the-shelf 3D face reconstruction model to extract the corresponding FLAME parameters and then apply average pooling, resulting a fused FLAME mesh $M(\boldsymbol{\beta_0}, \boldsymbol{\theta_0}, \boldsymbol{\psi_0})$ with 5023 vertices and 9976 faces. We further follow HeadStudio (Zhou et al., 2024) to increase the number of 3D Gaussian kernels for faster convergence. Specifically, we randomly sample 4 points on every face of the mesh, comprising approximately 40,000 points. To keep the geometry in a decent human head shape during optimization, we add the geometry constraint between the 3D portrait and the template FLAME mesh by associating every 3D Gaussian with its initial located triangle, assuming the optimized 3D Gaussian clusters do not undergo severe displacement from the geometry prior.

**Residual Gaussian Derivation.** During training, we follow the common practice of head avatar (Shao et al., 2024; Hu et al., 2024) to optimize the Gaussian splats with a residual scheme based on the FLAME model. Specifically, we formulate the 3D portrait Gaussian as a combination of a FLAME-embedded triangular Gaussians and a residual Gaussian term as the offset. Instead of learning the Gaussian parameters from scratch directly, we derive the properties of triangular Gaussian from the template mesh deformed by a set of FLAME parameters $(\boldsymbol{\beta}, \boldsymbol{\theta}, \boldsymbol{\psi})$ via linear blend skinning process (Lewis et al., 2023). Given the flame mesh $M(\boldsymbol{\beta}, \boldsymbol{\theta}, \boldsymbol{\psi})$, each triangular face can be transformed as a Gaussian kernel $G_{tri}$ with $\{\mu_{tri}, R_{tri}, s_{tri}\}$ where $\mu_{tri}$ is the centroid of the triangle (the average position of three vertices), $R_{tri}$ is the rotation matrix of the triangle, and $s_{tri}$ is scaling matrix with scale factor as the average length of the base and height. During rendering, the $i$-th Gaussian kernel $G_i$ can be derived based on the transformed triangle Gaussian $G_{tri}$ and residual Gaussian term parameterized by $\{\Delta\mu_i, \Delta R_i, \Delta s_i\}$ as follows:

$$\mu_i = s_{tri} R_{tri} \Delta\mu_i + \mu_{tri}, \quad r_i = R_{tri} \Delta R_i, \quad s_i = s_{tri} \Delta r_i, \tag{6}$$

where $\mu_i$, $r_i$, and $s_i$ are the mean, rotation matrix, and scaling factor, respectively.

**Absolute Gradient Strategy.** To improve the performance of the novel view synthesis and overcome the over-blur problem, we adopt an absolute gradient sum strategy (Ye et al., 2024; Yu et al., 2024) to solve the gradient collision problem in densification. The original 3D GS calculates the positional gradient $\frac{\partial L}{\partial \mu_i^{2D}}$ to guide the densification:

$$\frac{\partial L}{\partial \mu_i^{2D}} = \left( \sum_j \frac{\partial L_j}{\partial \mu_{i,x}^{2D}}, \sum_j \frac{\partial L_j}{\partial \mu_{i,y}^{2D}} \right), \tag{7}$$

where $\mu_i^{2D} = (\mu_{i,x}^{2D}, \mu_{i,y}^{2D})$ is the center of projected 2D Gaussian, and $j$ indicates the $j$-th pixel contributed to the projected 2D Gaussian. It is worth noting that the sub-gradients use opposite signs to indicate different directions in the $x$ and $y$ axes. Consequently, the overall positional gradient magnitude $\frac{\partial L}{\partial \mu_i^{2D}}$ can be suppressed if the sub-gradients with opposite signs negate each other. To resist

the gradient collision issue, Instax3D accumulates the absolute value of every pixel sub-gradients with:

$$\frac{\partial L'}{\partial \mu_i^{2D}} = \left( \sum_j \left| \frac{\partial L_j}{\partial \mu_{i,x}^{2D}} \right|, \sum_j \left| \frac{\partial L_j}{\partial \mu_{i,y}^{2D}} \right| \right). \tag{8}$$

The absolute sum can aggregate each pixel's contribution by aggregating the magnitudes of sub-gradients in $x$ and $y$ dimensions.

## 4 EXPERIMENTS AND RESULTS

**Datasets.** To finetune the video diffusion module, we construct a dataset of 3D full-head videos. Due to the scarcity of the real-human 3D head scan dataset, we utilize a pre-trained 3D-ware generator Panohead (An et al., 2023) to produce $1,000$ synthetic turn-around videos. For every training video, we randomly draw a Gaussian noise to generate a tri-grid representation, and then render it into video frames from views with a fixed elevation and $K = 21$ evenly distributed horizontal angles. During training, we pick one frontal view as the image condition and the first frame of the generated video.

**Runtime.** The process of generating a 3D portrait using Instax3D necessitates 10 minutes on a single NVIDIA V100 GPU. This initial step of the turn-around video generation process can be completed in approximately 1 minute. The multi-view FLAME-fused initialization process takes around 3 minutes. The final and most computationally intensive step that reconstructs a 3D portrait consumes about 6 minutes.

**Evaluation Metrics.** We evaluate the effectiveness of the proposed Instax3D from these aspects: identity preservation, multi-view consistency, and shape and pose accuracy. For identity preservation, We use two pre-trained face recognition networks, *i.e.*, Arcface (Deng et al., 2019a) and Facenet (Schroff et al., 2015), to extract the facial identity feature from the image, and then calculate consistency metric between the rendered images and reference image. For the multi-view consistency, we use the average of CLIP (Radford et al., 2021) and DINO (Caron et al., 2021) scores through appearance similarity across the different views. Specifically, we adopt the open-sourced clip-vit-large-patch14[1] model and DeiT-S based DINO [2] as the feature extractors. To evaluate the correctness of generated 2D poses, we calculate the Percentage of Correct Keypoint (PCK) metric. To compute PCK, we use a face key point detection model, *i.e.*, MTCNN[3], to detect the facial key points on the synthesized images. Then, we calculate the percentage of detected key points with the ground-truth keypoint map projected from the 3D FLAME model. We further compute the mean squared error (MSE) of the shape and pose parameters between the ground truth and the generated 3D portraits. Here we leverage a 3D face detection model DECA (Feng et al., 2021) to estimate the 3D shape and pose code.

**Comparison Baselines.** We compare our Instax3D with different kinds of methods: SDS-based (Poole et al., 2022) optimized methods (Qian et al., 2024; Yu et al., 2023), large reconstruction models (Hong et al., 2024; Tang et al., 2024), and image prompt adapter models (Ye et al., 2023; Wang et al., 2024). Here we choose several representative baselines including Magic123 (Qian et al., 2024), ImageDream (Wang & Shi, 2023), Wonder3D (Long et al., 2024), LRM (Hong et al., 2024), IP-Adapter (Ye et al., 2023) and CapHuman (Liang et al., 2024). Following previous practices, we conduct experiments on $512 \times 512$ resolutions for a fair comparison. For Magic123 (Qian et al., 2024), we optimize with Stable Diffusion v1.5 pipeline with ControlNetMediaPipeFace[4] as the 2D diffusion prior and Zero-1-to-3 (Liu et al., 2023) as the 3D prior. In Wonder3D (Long et al., 2024), we first generate consistent multi-view normal maps and color images from 6 views, and then perform mesh extraction with Instant-NSR (Guo, 2022). To adapt IP-Adapter (Ye et al., 2023) to multi-view scenarios, we incorporate the Realistic Vision Lora module[5] with a ControlNet model, *i.e.*, ControlNetMediaPipeFace[6], to achieve the control over the human facial poses and expressions.

---

[1] https://huggingface.co/openai/clip-vit-large-patch14
[2] https://github.com/facebookresearch/dino.git
[3] https://github.com/timesler/facenet-pytorch
[4] https://huggingface.co/CrucibleAI/ControlNetMediaPipeFace
[5] https://huggingface.co/collections/SG161222
[6] https://huggingface.co/CrucibleAI/ControlNetMediaPipeFace

Table 1: **Quantitative comparison.** We compare Instax3D with several baseline methods. (1) SDS-based optimized method: Magic123 (Qian et al., 2024) and ImageDream (Wang & Shi, 2023) (2) Large reconstruction model methods: LRM (Hong et al., 2024) and LGM (Tang et al., 2024) (3) Image prompt methods: Wonder3D (Long et al., 2024), IP-Adapter Ye et al. (2023), and Caphuman (Liang et al., 2024). Here we use MVC and IPA to represent the multi-view consistency and identity preservation ability, respectively. PCK denotes the percentage of correct keypoint. We mark out best , second best , and third best metrics of single-view 3D protraits generation methods.

| Method | Publication | MVC ↑ | IPA ↑ | PCK ↑ | Shape ↓ | Pose ↓ |
|---|---|---|---|---|---|---|
| *SDS-based optimized methods* | | | | | | |
| Magic123 | ICLR 2024 | 0.567 | 0.268 | 60.7 | 0.251 | 0.078 |
| ImageDream | Arxiv | 0.704 | 0.720 | 69.4 | 0.329 | 0.065 |
| *Large reconstruction model methods* | | | | | | |
| LRM | ICLR 2024 | 0.542 | 0.318 | 70.2 | 0.468 | 0.075 |
| LGM | ECCV 2024 | 0.569 | 0.356 | 71.5 | 0.380 | 0.046 |
| *Image adapter methods* | | | | | | |
| Wonder3D | CVPR 2024 | 0.738 | 0.582 | 77.4 | 0.147 | 0.053 |
| IP-Adapter | Arxiv | 0.837 | 0.771 | 73.7 | 0.437 | 0.041 |
| CapHuman | CVPR 2024 | 0.717 | 0.748 | 87.4 | 0.232 | 0.038 |
| **Instax3D** | | 0.905 | 0.741 | 95.4 | 0.133 | 0.027 |

For Caphuman (Liang et al., 2024), we render the 3D Parametric Face Model Flame (Li et al., 2017) into the normal images as head conditions to achieve fine-grained head control. For LRM (Hong et al., 2024), we adopt the OpenLRM[7] implementation (He & Wang, 2023), and render the human head with a fixed camera intrinsics and extrinsics.

### 4.1 EXPERIMENTAL RESULTS

**Quantitative Evaluations.** We further show the evaluation results in Table 1. According to the framework, these methods can be categorized into SDS-based (Poole et al., 2022) optimized methods (Qian et al., 2024; Yu et al., 2023), large reconstruction models (Hong et al., 2024; Tang et al., 2024), and image prompt adapter models (Ye et al., 2023; Wang et al., 2024). We compare these methods from these aspects: multi-view consistency, identity preservation ability, shape correctness and pose controllability. Our approach outperforms all other methods in all the metrics. We observe the SDS-based optimized methods struggle to maintain the multi-view consistency and have a low score in the PCK metric (percentage of correct key points). The image adapter methods achieve relatively high scores in the identity preservation metric but sacrifice the pose controllability (PCK) for better identity preservation. The large reconstruction model methods suffer from bad multi-view consistency. Benefiting from the inherent multi-view knowledge from the video diffusion module, Instax3D demonstrates strong abilities in maintaining strong multi-view consistency across different viewpoints. The high IPA score proves the combination of VAE and CLIP (Radford et al., 2021) encoders can capture image features effectively. In terms of the metrics of 3D shape and pose, our method also surpasses all other methods, which can be attributed to the FLAME-based initialization and residual Gaussian design.

**Qualitative Comparison.** Given reference images and target poses, we present a qualitative comparison against several competitive baselines by visualizing the rendered novel view results of the generated 3D portraits. As shown in Figure 2, we can make the following observations. Firstly, IP-Adapter (Ye et al., 2023), Wonder3D (Long et al., 2024), and Caphuman (Liang et al., 2024) exhibit the typical Janus problem when synthesizing back-view image (generating the front face image in the back-view). LGM (Tang et al., 2024) and LRM (Hong et al., 2024) fail to learn a reasonable 3D shape and suffer from the blurry and "cloud-like" artifacts in the back of the head. 2D-lifting techniques such as Magic123 (Qian et al., 2024) and Wonder3D (Long et al., 2024) produce a flat human head. The "plate-like" effect can be attributed to the lack of 3D-awareness and multi-view knowledge, because 2D diffusion models only provide single-view supervision. Secondly, for iden-

---

[7] https://github.com/3DTopia/OpenLRM

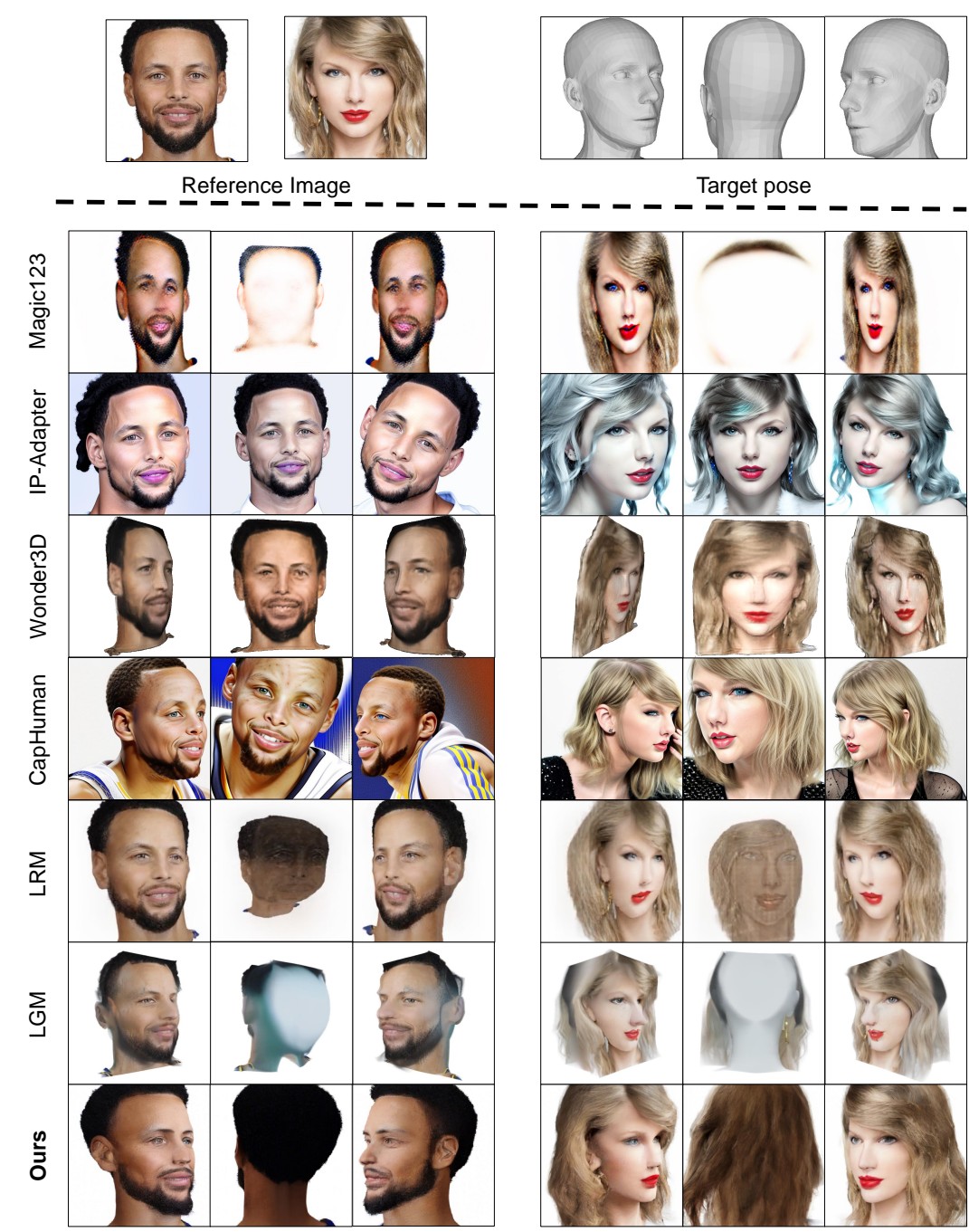

Figure 2: Qualitative comparisons with six baselines. Compared with other methods, Instax3D generates 3D portraits with various head poses while capturing fine-grained identity details. For video comparison, please refer to project page.

tity preservation, we find that the IP-Adapter (Ye et al., 2023) can only preserve part content of the facial region while changing other details such as the color of lips and hair. Caphuman (Liang et al., 2024) produce novel view results with cartoon-like visuals with vibrant colors, and add extra clothing. Thirdly, we observe that the IP-adapter (Ye et al., 2023) fails to control the head pose precisely when synthesizing the non-frontal pose images (see the 2-*rd* row of the Figure 2). Caphuman (Liang

Table 2: **User study.** We conduct user studies to assess the generation quality quantitatively. We also report the time cost for one 3D portrait and the frame-per-second (FPS) for rendering.

| Method | Publication | Quality ↑ | Alignment ↑ | Time ↓ | FPS ↑ |
|---|---|---|---|---|---|
| *GAN inversion methods* | | | | | |
| Portrait3D | SIGGRAPH 2024 | 3.4 | 3.0 | ∼1.5 hours | ∼ 3 |
| *Image adapter methods* | | | | | |
| Wonder3D | CVPR 2024 | 3.1 | 2.6 | ∼5 minutes | ∼ 50 |
| *SDS-based optimized methods* | | | | | |
| Magic123 | ICLR 2024 | 2.1 | 3.5 | ∼ 1 hour | ∼ 20 |
| ImageDream | Arxiv | 2.9 | 2.3 | ∼ 2 hours | ∼ 5 |
| *Large reconstruction model methods* | | | | | |
| LRM | ICLR 2024 | 3.2 | 3.4 | ∼ 5 seconds | ∼ 70 |
| LGM | ECCV 2024 | 3.6 | 4.0 | ∼ 5 seconds | ∼ 60 |
| **Instax3D** | | 3.9 | 4.2 | ∼ 10 minutes | ∼ 40 |

et al., 2024) can generate side-profile images but suffers from incorrect results in the back view of the head. In comparison with 2D-lifting techniques (Qian et al., 2024; Ye et al., 2023; Long et al., 2024; Liang et al., 2024), our Instax3D generates 3D portraits with better 3D awareness and multi-view consistency, benefit from the video diffusion prior. Besides, thanks to its explicit geometry prior and the 3D head modeling, Instax3D also outperforms the large reconstruction models with better 3D shapes.

**User Study.** We further conduct user studies to assess the generation quality of Instax3D by comparing against four image-to-3D methods. We pick 10 human images to create corresponding 3D portraits, which were then rendered into turn-around videos for visualization. For each reference image, we generate the rendered videos via different methods and collect feedback from volunteers regarding image quality and alignment with the reference image. We get 200 responses from 20 participants in total. From Table 2, we observe that Instax3D obtains superior preference over all other methods. We also add the time cost required for generating a 3D portrait. The proposed method can create a trade-off between time cost and overall quality.

**Abaltion Studies** We further conduct ablation studies to explore the impact of different designs. Please refer to the Appendix for the experimental results.

## 5 CONCLUSION

This work presents Instax3D, a fast 3D portrait creation solution for generating 3D full-head Gaussians from a reference image within 10 minutes. The 3D portrait creation problem is divided into "generation and construction" process. In the generation stage, Instax3D deploys a fine-tuned image-to-video generation model to imagine the novel view results of the given single-view image, harnessing the inherent multi-view consistency and strong 3D awareness of the video diffusion model. The double encoder design, *i.e.*, CLIP and VAE, can extract the fine-grained details effectively, ensuring the strong identity preservation ability. In the construction stage, we construct the 3D portrait with a multi-view FLAME-based 3D Gaussian splitting representation, harnessing both the fast converging abilities of 3D Gaussian Splatting and the geometric guidance of the expressive FLAME model. Experimental results demonstrate that the proposed method can make a great trade-off between the time cost and the overall quality.

**Limitations and future work.** One limitation is that it lacks the character animation ability to deform the generated 3D portraits to novel poses and shapes. In this paper, we mainly focus on the scenarios of a static human character. To adapt Instax3D to the animatable portrait scenarios, one possible way is to integrate a pose guider into the video diffusion model to drive the movements of the character (Tian et al., 2024; Xu et al., 2024). Another limitation is that the total process still costs several minutes for every 3D asset. In the future, we plan to explore the feed-forward inference-only methods with large 3D generation models.

**Ethics Statement.** Our 3D portrait creation method Instax3D is built upon a video generation model. Therefore, our model inherits both the capabilities and limitations of these foundational diffusion models, and thus might introduce several ethical considerations. Our approach could potentially be misused to generate inappropriate content such as fake portrait creation. Therefore, we believe that any images or models produced using the proposed method should undergo a thorough review and be clearly labeled as synthetic. We are dedicated to ensuring that our work complies with legal standards, especially regarding intellectual property, data privacy, and the ethical implications of video generation technologies.

**Reproducibility Statement.** Our Instax3D is built publicly available codebases, *i.e.*, generative-models[8], gaussian-splatting[9], and AbsGS[10]. We also include the data pre-processing details, implementation details in Section 4 and Appendix, and to facilitate reproducing Instax3D.

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

In the appendix, we first provide more implementation details in Section A. Second, we conduct the ablation studies in Section B. Finally, we show more visualization results in Section C. Please refer to our project page for video results.

# A  IMPLEMENTATION DETAILS

## A.1  TRAINING PROTOCOL

In this work, we use SV3D (Voleti et al., 2024) as the video diffusion model. To facilitate the training process, we preprocess the video data by using the VAE encoder from Stable Video Diffusion (SVD) (Blattmann et al., 2023a) to encode the input image as a video latent. During training, we freeze the Clip encoder and fine-tune the U-Net block with the learning rate $1 \times 10^{-5}$ for 4000 steps. We use AdamW optimizer Loshchilov & Hutter (2019) with $\beta_1 = 0.9$, $\beta_2 = 0.999$, and the batch size of 1 for fine-tuning. During inference, we adopt DDIM sampler (Song et al., 2021a) using triangular classifier-free guidance. We finetune the video diffusion model on a Nvidia A40 GPU with 46GB memory.

In the construction stage, we employ Deep3DFaceReconstruction (Deng et al., 2019b) and Faceverse (Wang et al., 2022) to estimate the camera pose and the gaze direction, respectively, and then regress the FLAME parameter (Li et al., 2017) with an off-the-shelf face detector DECA (Feng et al., 2021). During the 3D Gaussian optimization process, all 21 video frames generated from the fine-tuned video diffusion model are used to reconstruct the 3D human portraits. We use Adam optimizer Kingma & Ba (2015) with $\beta_1 = 0.9$, $\beta_2 = 0.999$ for optimization. We optimize the Gaussian Splatting representation for 20,000 iterations, applying an exponential decay to the learning rate for splat positions until it reaches 0.01× the initial value at the final iteration. We perform adaptive density control with absolute gradient sum strategy (Ye et al., 2024; Yu et al., 2024) every 1,000 steps, and the gradient threshold for densification is set as 0.0002.

## A.2  DATASETS

In the evaluation part, we conduct quantitative comparison experiments on a subset from a large-scale in-the-wild human face dataset, *i.e.*, Flickr-Faces-HQ (FFHQ)[11] (Karras et al., 2019) dataset. FFHQ is known for its various styles and extensive diversity, encompassing a wide range of ethnicities, ages, and image backgrounds, along with substantial variations in facial attributes and accessories such as hats, eyeglasses, and earrings. Specifically, we pick 100 portrait images to conduct quantitative comparison experiments and ablation studies.

# B  ABLATION STUDIES

**FLAME guidance.** Instax3D adopts a FLAME-guided residual learning scheme to form the final full-head portrait on top of the triangular Gaussians derived from the coarse FLAME head mesh. To investigate the effectiveness of the Flame guidance, we compare our method's performance when reconstructing the 3D portrait with and without Flame prior. Here we report the shape and pose accuracy by computing the MSE in $10^{-2}$ of shape and parameters between the reference and the generated images. When removing the FLAME guidance, the shape error increases from 0.133 to 0.245, while the pose accuracy increases from 0.027 to 0.049. The results suggest that FLAME guidance is essential to preserve the 3D properties of the reconstructed portraits, *i.e.*, 3D shape and head pose.

**Camera Modulation.** We further conduct experiments to explore different camera modulations, *i.e.*, camera-aware and camera-invariant conditions. As shown in the left part of Figure. 3, we observe Instax3D can generate distorted and collapsed results in the novel views when using camera embedding as the conditions (camera-aware modulation). The generation results is very sensitive to the sensitive to the camera parameters, *e.g.*, intrinsics and extrinsics, elevation and azimuth angles. We conjecture that the camera embeddings serve as a strong prior in the denoising process, and thus are likely to guide the video diffusion model to produce averaged results with the same condition. Besides, there is a significant gap in camera distribution between the training data and the testing

---

[11] https://github.com/NVlabs/ffhq-dataset

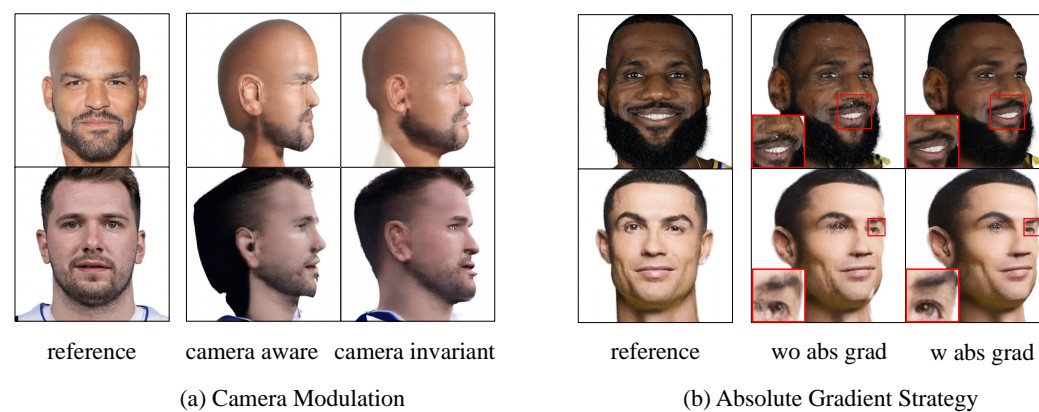

(a) Camera Modulation                    (b) Absolute Gradient Strategy

Figure 3: Abatlion on different model designs.

data. Any slight perturbation caused by the inaccurate estimation of the camera parameters can lead to de-generated results. In comparison, we find that using the camera-invariant conditions (zero embedding) leads to more robust results.

**Absolute gradient strategy.** To assess the efficacy absolute gradient strategy, we compare the performance of our Instax3D when optimized with the default adaptive density control and the absolute gradient strategy. The qualitative comparison is shown in the right part of Figure 3. We observe the default densification strategy fails to identify the well-optimized and over-blur regions. In contrast, the absolute gradient strategy can mitigate the over-blurred issue and produce a clearer appearance for the created 3D portrait.

# C  ADDITIONAL RESULTS

Please refer to our project page for video results.

