# OpenReview forum: "Instax3D: Creating 3D Portrait from a single-view image in Minutes"
_ICLR.cc/2025/Conference — ICLR 2025 Conference Withdrawn Submission_

### Official Review · Reviewer_x4C1 · 2024-10-16

**Soundness:** 2
**Presentation:** 3
**Contribution:** 1
**Rating:** 3
**Confidence:** 5

**Summary:**

The paper proposed a novel method on single-view 3D portrait creation. It combines the priori from video diffusion model and model-based(FLAME) 3D guidance. With specific designed strategies(like Multi-view FLAME-fused Initialization and Residual Gaussian Derivation). It achieves comparable or better results compared with SoTA image-to-3D methods(both quantitative and qualitative).

**Strengths:**

1、The author  clearly present the core idea and experimental settings.
2、The numerical metric and visualization is competitive among SoTA image-to-3D methods.
3、Overall，the method is novel in formulation but not in new designed blocks for the task. So I have some concerns, see Weaknesses.

**Weaknesses:**

1、Data is the upper bound of any algorithms. The author says that they utilize a pre-trained 3D-ware generator
Panohead (An et al., 2023) to produce 1, 000 synthetic turn-around videos for training. We know that the turning around videos generated by 3D-Aware GAN-based methods sometimes may not be that 3D-consistent. I wonder whether the author manually filter out those samples which is distorted? Was 1000 ids sufficient for algorithm training to make it generalize well? I'd like to see more cases.

2、Image-to-3D methods like Wonder3D、ImageDream was not trained on specific human head dataset. It will be more fair to finetune them on your specific dataset.

3、Some talking head methods can also turn around the face, and drive the face with any facial animation. However, the proposed method can only produce a static head model. I can not figure out the strength of the proposal when compared with the following works. Since in most application cases, we only need to see the front face of the virtual portrait.
1) Real3D-Portrait: One-shot Realistic 3D Talking Portrait Synthesis
2) X-Portrait: Expressive Portrait Animation with Hierarchical Motion Attention

**Questions:**

1、The whole pipeline is novel but the author do not propose a specific novel module designed for the task if I do not neglect something important. The author could explain this further.
2、Strengthes of the paper compared with talking head methods. I'd like to see the author's opinions.
3、Why use synthetic turn-around videos？I'd like to see the author's opinions.

---

### Official Review · Reviewer_2ZgQ · 2024-10-30

**Soundness:** 3
**Presentation:** 3
**Contribution:** 2
**Rating:** 3
**Confidence:** 4

**Summary:**

This paper proposes a method for reconstructing 3D portraits from single-view images. The authors first fine-tune SV3D, a multi-view video generator, on a portrait multi-view dataset, and then utilize 3D Gaussian Splatting to perform the reconstruction from this multi-view data. Notably, the paper also introduces FLAME-based 3D-GS, and incorporates a residual neural deviation to enhance the optimization of the 3D representation.

**Strengths:**

+ This paper addresses the intriguing challenge of generating 3D portraits from single-view images, proposes a well-structured pipeline, fine-tunes a multi-view diffusion model using a specialized dataset.
+ The writing is clear, facilitating an understanding of the technical details without significant effort.

**Weaknesses:**

+ Novelty of the Pipeline: This paper presents a "generation-then-reconstruction" pipeline for single-view 3D portrait generation. However, several existing works, such as CAT3D [1], ReconX [2], ViewCrafter [3], and MagicMan [4], have explored similar schemes across various domains, including single-view 3D object, scene, and human generation. Furthermore, initializing with a parametric mesh is a common practice for 3D Gaussian initialization. For instance, methods like HumanGaussian [5] and DreamWaltz [6] use the SMPL-X template for this purpose.

+ Unsatisfactory Quantitative Results and Incomplete Baselines:
The authors selected baselines such as LGM, LRM, ImageDream, IPAdapter, and Magic123, but overlooked critical baselines relevant to human head avatar generation. Both feed-forward methods (LRM/LGM) and multi-view generators (Wonder3D, ImageDream, Magic123) were initially designed for general 3D object generation, relying on Objaverse for training.
Finetuning these models on the portrait dataset is essential if the authors intend to compare their method against these general-purpose approaches. Additionally, for portrait generation, Rodin [7] and RodinHD [8] could serve as valuable baselines.
The qualitative results presented in the video demo do not appear to match the quality of RodinHD.

+ More: I do not believe the pixel-wise optimization formulation is the main reason for efficiency.
Moreover, SDS-based methods are not inherently slow if the 3D representation is built on 3D-GS, as evidenced by GaussianDreamer [9] and DreamGaussian [10].
A major concern is the quality associated with SDS-based methods, particularly issues like texture oversaturation.
The challenge of reconstructing from generated multi-view images is also noteworthy, given the potential for multi-view inconsistencies.

[1] Gao et al., CAT3D: Create Anything in 3D with Multi-View Diffusion Models, arXiv 2024.

[2] Liu et al., ReconX: Reconstruct Any Scene from Sparse Views with Video Diffusion Model, arXiv 2024.

[3] Viewcrafter: Taming video diffusion models for high-fidelity novel view synthesis, arXiv 2024.

[4] MagicMan: Generative Novel View Synthesis of Humans with 3D-Aware Diffusion and Iterative Refinement, arXiv 2024.

[5] Liu et al., Text-Driven 3D Human Generation with Gaussian Splatting, CVPR 2024.

[6] Huang et al., Make a Scene with Complex 3D Animatable Avatars, NeurIPS 2023.

[7] Wang et al., Rodin: A Generative Model for Sculpting 3D Digital Avatars Using Diffusion, CVPR 2023.

[8] Zhang et al., High-Fidelity 3D Avatar Generation with Diffusion Models, ECCV 2024.

[9] Yi et al., GaussianDreamer: Fast Generation from Text to 3D Gaussians by Bridging 2D and 3D Diffusion Models, CVPR 2024.

[10] Tang et al., DreamGaussian: Generative Gaussian Splatting for Efficient 3D Content Creation, ICLR 2024.

**Questions:**

+ The term "reconstruction" may be more appropriate than "construction" in the context of obtaining the 3D representation from multi-view images (or video).

+ SDS-based methods are not necessarily slow to optimize. In fact, they can mitigate some drawbacks associated with pixel-level optimization by operating on the feature level for 3D representation. I encourage the authors to explore incorporating SDS-based methods (or hybrid possibly) and to discuss this aspect in greater depth.

+ Regarding novel view synthesis metrics: The multi-view inconsistency and pixel-wise reconstruction challenges can lead to unsatisfactory results, such as blurriness. Can the authors elaborate on this issue and provide a quantitative evaluation of the reconstruction quality?

---

### Official Review · Reviewer_3ybj · 2024-11-01

**Soundness:** 3
**Presentation:** 3
**Contribution:** 1
**Rating:** 3
**Confidence:** 5

**Summary:**

The author presents a framework, namely Instax3D, which generates a 3D portrait from a single-view image. This pipeline consists of two stages: 1) Multi-view Generation for Multi-view head images and 2) Multi-view Reconstruction. As the experiments demonstrate, their results are competitive compared to prior work.

**Strengths:**

The paper is well-written and easy to follow.

**Weaknesses:**

1) Although the authors wish to generate multi-view images from a video diffusion model and subsequently apply a multi-view reconstruction method to create a static human avatar from the generated images, there is a core issue that they do not address. The main problem is that the images produced by the multi-view generation model are view inconsistent. How can one reconstruct a static Gaussian portrait from multi-view inconsistent images? More importantly, what do the outputs of your video diffusion model look like? Are the rendering results of your 3D Gaussian model overfitting the video image output? Please show me more novel view synthesis results.


2) The novelty is limited. The author straightforwardly fine-tunes a video diffusion model using the generated head model. This strategy is somewhat Uncanny. Does it imply that the upper bound of the generated multi-view image is the rendered view of the generated portrait model? Additionally, we could not find any interesting designs in the multi-view 3DGS reconstruction.


3) Their results do not capture my attention. Honestly, the quality of the generated heads is not satisfactory.

4) The comparison results are unfair. The authors employ a pre-trained 3D-aware generator to produce 1,000 synthetic self-rotating videos to fine-tune their multi-view video model. For a fairer comparison, I suggest that the authors fine-tune the authors' baseline models, such as Wonder3D and LGM, to obtain their results instead of simply resuming their weights to generate their results.

**Questions:**

No other questions.

---

### Official Review · Reviewer_wHjd · 2024-11-04

**Soundness:** 2
**Presentation:** 2
**Contribution:** 2
**Rating:** 3
**Confidence:** 4

**Summary:**

The paper introduces Instax3D, a framework for quickly generating photorealistic and identity-preserving 3D portraits from a single headshot. Utilizing a generative Gaussian Splatting model with a video diffusion prior, it synthesizes high-quality 3D portraits in minutes. The process consists of a generation stage, where a fine-tuned video diffusion model creates a multi-view video sequence, and a construction stage that reconstructs the 3D portrait using 3D Gaussian Splatting and the FLAME morphable face model.

**Strengths:**

The paper has a clear writing structure and a well-defined motivation. Additionally, the experimental section effectively demonstrates the validity of the proposed method. The paper also involves significant engineering efforts, as it includes fine-tuning SVD, implementing flame estimation based on multi-view images, and performing 3D reconstruction.

**Weaknesses:**

The paper has the following shortcomings:
Effectiveness: While it is true that the proposed method outperforms techniques like LRM in image reconstruction, it remains unclear whether the improved results stem from the use of SVD rather than the reconstruction algorithm itself.

Shape and Texture Issues: Observations from the 360-view rendering on the webpage show that the side-view renderings have significant shape inaccuracies. Additionally, the textures in novel views lack realism.

Lack of Comparisons: The paper does not include comparisons with 3D GAN-based methods such as EG3D, Next3D, LP3D(Live 3D Portrait), and Panohead, which could provide valuable context for its findings.

**Questions:**

1. If LRM and LGM were provided with more consistent multi-view images, would their outcomes also be enhanced?
2. The paper says that it uses Panohead to generate 360 full-head videos as dataset to finetune SVD. So why don't directly use Panohead for the image-to-3D task? What's the benefit of generating a video and then performing reconstruction over directly performiing GAN inversion with Panohead?

---

### Official Review · Reviewer_T8ah · 2024-11-06

**Soundness:** 3
**Presentation:** 3
**Contribution:** 4
**Rating:** 8
**Confidence:** 4

**Summary:**

This paper addresses the problem of reconstructing a 3D portrait from a single-view image, specifically creating a complete 3D head model. The authors propose Instax3D, a method that operates in three primary stages. First, it takes a single protrait image and, using a fine-tuned video diffusion model, generates a multi-view 360-degree selfie, capturing different angles around the head. Next, the model leverages this sequence to estimate a FLAME-based 3D head model. The FLAME model serves as a structural guide for initializing 3D Gaussians. Finally, the method optimizes these Gaussians based on the generated multi-view video, leading to a full 3D head reconstruction. Comparative experiments demonstrate Instax3D’s advantages over existing methods, particularly its ability to accurately reconstruct the head's rear view, and avoiding the Janus problem often seen in single-image-based approaches. Ablation studies also validate the efficacy of each component in the pipeline.

**Strengths:**

- Capture setup

3D reconstruction from a single portrait image is ambiguous due to the limited perspective. To address this challenge, Instax3D employs a video diffusion model to generate a multi-view video sequence from a single input image. This approach is novel as it effectively simulates a multi-view capture setup without the need for multiple cameras or images. By generating consistent and coherent views from various angles, Instax3D creates a rich, synthetic multi-view dataset that provides the information needed for 3D reconstruction. This setup not only ensures that multi-view consistency is maintained but also allows the method to inherit identity and 3D priors embedded within the video diffusion model. Such approach to 3D data collection has the potential to benefit the broader 3D reconstruction and generation community.

- Flame model

The use of the FLAME model as a guide for 3D reconstruction is a strength of Instax3D. Instax3D leverages FLAME to initialize the 3D Gaussians, inheriting a strong 3D prior from this morphable model. As a result, it addresses the Janus problem and can successfully synthesize the rear view of the head. It also effectively handles view synthesis when the viewing angle differs significantly from the input view. This approach could inspire the community by demonstrating a way to directly inject strong 3D priors into the generative models, even in a single-view image setting.

- Residual Gaussian Derivation

Instax3D introduces Residual Gaussian Derivation to enhance the optimization process by aligning the Gaussians closely with the FLAME model. This residual learning scheme enables the 3D reconstruction to closely follow the guidance of the FLAME mesh, which serves as a proxy for the head’s geometry. It optimizes only for location, rotation, and scale offsets, ensuring that the optimization does not deviate too far from the FLAME initialization. This well-designed optimization scheme is effective in making the 3D Gaussians adhere to an explicit 3D model.

Overall the paper is well-written and easy to follow.

**Weaknesses:**

- Video Diffusion Model

The quality of the 3D reconstruction produced by Instax3D is inherently limited by the quality of the synthesized 360-degree selfie generated by the video diffusion model. Any artifacts or inaccuracies in this initial synthetic sequence are carried over into the final 3D reconstruction, which may impact the realism and fidelity of the result. While the authors claim that the 3D reconstruction benefits from the identity and shape priors of the video diffusion model, they do not provide any examples of the synthesized 360-degree video. Including sample views for each column shown in Figure 2 would better illustrate the effectiveness of these priors and the quality of the synthetic sequence generated.

- Comparisons

In the evaluation section, the paper primarily compares Instax3D with recent generative models for 3D generation but lacks comparisons with methods focused specifically on portrait reconstruction. Given that the video generation model produces a 360-degree video, Instax3D could also be evaluated against fully optimization-based models or fine-tuned 3D GAN-based methods for synthesizing novel views. Including these comparisons would provide a clearer perspective on Instax3D’s relative performance and help determine whether its major performance gains come from the synthesized 360 video or from the 3D Gaussian reconstruction.

- Convexity in side-views

In several examples shown on the project page, the convexity of the reconstructed head appears inaccurate, especially in side views where the face can look concave rather than convex. This issue is not present in the FLAME model, suggesting that it may come from the 3D Gaussian optimization process. To better understand the source of this issue, it would be helpful to include side-view portraits from the synthesized 360-degree video, providing additional insight into whether the problem originates in the video generation or during the optimization stage.

**Questions:**

It would be helpful to list the training time and FPS in Table 1.

L152 guarantee the *positive* semi-definite property.

**Details Of Ethics Concerns:**

Face reconstruction method could result in misuses such as deep fake.

---

### Note · Authors · 2024-11-14

I have read and agree with the venue's withdrawal policy on behalf of myself and my co-authors.